# Statistical and Computational Trade-Offs in Kernel K-Means

**Daniele Calandriello**
LCSL – IIT & MIT,
Genoa, Italy

**Lorenzo Rosasco**
University of Genoa,
LCSL – IIT & MIT

## Abstract

We investigate the efficiency of k-means in terms of both statistical and computational requirements. More precisely, we study a Nyström approach to kernel k-means. We analyze the statistical properties of the proposed method and show that it achieves the same accuracy of exact kernel k-means with only a fraction of computations. Indeed, we prove under basic assumptions that sampling $\sqrt{n}$ Nyström landmarks allows to greatly reduce computational costs without incurring in any loss of accuracy. To the best of our knowledge this is the first result of this kind for unsupervised learning.

## 1 Introduction

Modern applications require machine learning algorithms to be accurate as well as computationally efficient, since data-sets are increasing in size and dimensions. Understanding the interplay and trade-offs between statistical and computational requirements is then crucial [31, 30]. In this paper, we consider this question in the context of clustering, considering a popular nonparametric approach, namely kernel k-means [33]. K-means is arguably one of most common approaches to clustering and produces clusters with piece-wise linear boundaries. Its kernel version allows to consider nonlinear boundaries, greatly improving the flexibility of the approach. Its statistical properties have been studied [15, 24, 10] and from a computational point of view it requires manipulating an empirical kernel matrix. As for other kernel methods, this latter operation becomes unfeasible for large scale problems and deriving approximate computations is subject of a large body of recent works, see for example [34, 16, 29, 35, 25] and reference therein.

In this paper we are interested into quantifying the statistical effect of computational approximations. Arguably one could expect the latter to induce some loss of accuracy. In fact, we prove that, perhaps surprisingly, there are favorable regimes where it is possible maintain optimal statistical accuracy while significantly reducing computational costs. While a similar phenomenon has been recently shown in supervised learning [31, 30, 12], we are not aware of similar results for other learning tasks.

Our approach is based on considering a Nyström approach to kernel k-means based on sampling a subset of training set points (landmarks) that can be used to approximate the kernel matrix [3, 34, 13, 14, 35, 25]. While there is a vast literature on the properties of Nyström methods for kernel approximations [25, 3], experience from supervised learning show that better results can be derived focusing on the task of interest, see discussion in [7]. The properties of Nyström approximations for k-means has been recently studied in [35, 25]. Here they focus only on the computational aspect of the problem, and provide fast methods that achieve an *empirical* cost only a multiplicative factor larger than the optimal one.

Our analysis is aimed at combining both statistical *and* computational results. Towards this end we derive a novel *additive* bound on the empirical cost that can be used to bound the true object of interest: the *expected cost*. This result can be combined with probabilistic results to show that optimal statistical accuracy can be obtained considering only $O(\sqrt{n})$ Nyström landmark points,

where $n$ is the number of training set of points. Moreover, we show similar bounds not only for the optimal solution, which is hard to compute in general, but also for approximate solutions that can be computed efficiently using $k$-means++. From a computational point of view this leads to massive improvements reducing the memory complexity from $O(n^2)$ to $O(n\sqrt{n})$. Experimental results on large scale data-sets confirm and illustrate our findings.

The rest of the paper is organized as follows. We first overview kernel $k$-means, and introduce our approximate kernel $k$-means approach based on Nyström embeddings. We then prove our statistical and computational guarantees and empirically validate them. Finally, we present some limits of our analysis, and open questions.

## 2 Background

**Notation** Given an input space $\mathcal{X}$, a sampling distribution $\mu$, and $n$ samples $\{\mathbf{x}_i\}_{i=1}^n$ drawn i.i.d. from $\mu$, we denote with $\mu_n(A) = (1/n)\sum_{i=1}^n \mathbb{I}\{\mathbf{x}_i \in A\}$ the *empirical* distribution. Once the data has been sampled, we use the *feature map* $\varphi(\cdot) : \mathcal{X} \to \mathcal{H}$ to maps $\mathcal{X}$ into a Reproducing Kernel Hilbert Space (RKHS) $\mathcal{H}$ [32], that we assume separable, such that for any $\mathbf{x} \in \mathcal{X}$ we have $\phi = \varphi(\mathbf{x})$. Intuitively, in the rest of the paper the reader can assume that $\phi \in \mathbb{R}^D$ with $D \gg n$ or even infinite. Using the kernel trick [2] we also know that $\phi^\mathsf{T}\phi' = \mathcal{K}(\mathbf{x}, \mathbf{x}')$, where $\mathcal{K}$ is the kernel function associated with $\mathcal{H}$ and $\phi^\mathsf{T}\phi' = \langle \phi, \phi'\rangle_{\mathcal{H}}$ is a short-hand for the inner product in $\mathcal{H}$. With a slight abuse of notation we will also define the norm $\|\phi\|^2 = \phi^\mathsf{T}\phi$, and assume that $\|\varphi(\mathbf{x})\|^2 \leq \kappa^2$ for any $\mathbf{x} \in \mathcal{X}$. Using $\phi_i = \varphi(\mathbf{x}_i)$, we denote with $\mathcal{D} = \{\phi_i\}_{i=1}^n$ the input dataset. We also represent the dataset as the map $\Phi_n = [\phi_1, \dots, \phi_n] : \mathbb{R}^n \to \mathcal{H}$ with $\phi_i$ as its $i$-th column. We denote with $\mathbf{K}_n = \Phi_n^\mathsf{T}\Phi_n$ the empirical kernel matrix with entries $[\mathbf{K}_n]_{i,j} = k_{i,j}$. Finally, given $\Phi_n$ we denote as $\boldsymbol{\Pi}_n = \Phi_n\Phi_n^\mathsf{T}(\Phi_n\Phi_n^\mathsf{T})^+$ the orthogonal projection matrix on the span $\mathcal{H}_n = \mathrm{Im}(\Phi_n)$ of the dataset.

$k$**-mean's objective** Given our dataset, we are interested in partitioning it into $k$ disjoint *clusters* each characterized by its *centroid* $\mathbf{c}_j$. The Voronoi cell associated with a centroid $\mathbf{c}_j$ is defined as the set $\mathcal{C}_j := \{i : j = \arg\min_{s=[k]} \|\phi_i - \mathbf{c}_s\|^2\}$, or in other words a point $\phi_i \in \mathcal{D}$ belongs to the $j$-th cluster if $\mathbf{c}_j$ is its closest centroid. Let $\mathbf{C} = [\mathbf{c}_1, \dots \mathbf{c}_k]$ be a collection of $k$ centroids from $\mathcal{H}$. We can now formalize the criterion we use to measure clustering quality.

**Definition 1.** *The empirical and expected squared norm criterion are defined as*

$$W(\mathbf{C}, \mu_n) := \frac{1}{n}\sum_{i=1}^n \min_{j=1,\dots,k} \|\phi_i - \mathbf{c}_j\|^2, \qquad W(\mathbf{C}, \mu) := \mathop{\mathbb{E}}_{\phi \sim \mu}\left[\min_{j=1,\dots,k} \|\phi - \mathbf{c}_j\|^2\right].$$

*The empirical risk minimizer (ERM) is defined as* $\mathbf{C}_n := \arg\min_{\mathbf{C} \in \mathcal{H}^k} W(\mathbf{C}, \mu_n)$.

The sub-script $n$ in $\mathbf{C}_n$ indicates that it minimizes $W(\mathbf{C}, \mu_n)$ for the $n$ samples in $\mathcal{D}$. Biau et al. [10] gives us a bound on the excess risk of the empirical risk minimizer.

**Proposition 1** ([10])**.** *The excess risk $\mathcal{E}(\mathbf{C}_n)$ of the empirical risk minimizer $\mathbf{C}_n$ satisfies*

$$\mathcal{E}(\mathbf{C}_n) := \mathbb{E}_{\mathcal{D}\sim\mu}[W(\mathbf{C}_n, \mu)] - W^*(\mu) \leq \mathcal{O}\left(k/\sqrt{n}\right)$$

*where $W^*(\mu) := \inf_{\mathbf{C} \in \mathcal{H}^k} W(\mathbf{C}, \mu)$ is the optimal clustering risk.*

From a theoretical perspective, this result is only $\sqrt{k}$ times larger than a corresponding $\mathcal{O}(\sqrt{k/n})$ lower bound [18], and therefore shows that the ERM $\mathbf{C}_n$ achieve an excess risk optimal in $n$. From a computational perspective, Definition 1 cannot be directly used to compute $\mathbf{C}_n$, since the points $\phi_i$ in $\mathcal{H}$ cannot be directly represented. Nonetheless, due to properties of the squared norm criterion, each $\mathbf{c}_j \in \mathbf{C}_n$ must be the mean of all $\phi_i$ associated with that center, i.e., $\mathbf{C}_n$ belongs to $\mathcal{H}_n$. Therefore, it can be explicitly represented as a sum of the $\phi_i$ points included in the $j$-th cluster, i.e., all the points in the $j$-th Voronoi cell $\mathcal{C}_j$. Let $\mathcal{V}$ be the space of all possible disjoint partitions $[\mathcal{C}_1, \dots, \mathcal{C}_j]$. We can use this fact, together with the kernel trick, to reformulate the objective $W(\cdot, \mu_n)$.

**Proposition 2** ([17])**.** *We can rewrite the objective*

$$\min_{\mathbf{C} \in \mathcal{H}} W(\mathbf{C}, \mu_n) = \frac{1}{n} \min_{\mathcal{V}} \sum_{j=1}^k \sum_{i \in \mathcal{C}_j} \left\|\phi_i - \frac{1}{|\mathcal{C}_j|}\sum_{s \in \mathcal{C}_j} \phi_s\right\|^2$$

*with* $\left\|\phi_i - \frac{1}{|\mathcal{C}_j|}\phi_{\mathcal{C}_j}\mathbb{1}_{|\mathcal{C}_j|}\right\|^2 = k_{i,i} - \frac{2}{|\mathcal{C}_j|}\sum_{s \in \mathcal{C}_j} k_{i,s} + \frac{1}{|\mathcal{C}_j|^2}\sum_{s \in \mathcal{C}_j}\sum_{s' \in \mathcal{C}_j} k_{s,s'}$

While the combinatorial search over $\mathcal{V}$ can now be explicitly computed and optimized using the kernel matrix $\mathbf{K}_n$, it still remains highly inefficient to do so. In particular, simply constructing and storing $\mathbf{K}_n$ takes $\mathcal{O}(n^2)$ time and space and does not scale to large datasets.

# 3 Algorithm

A simple approach to reduce computational cost is to use approximate embeddings, which replace the map $\varphi(\cdot)$ and points $\phi_i = \varphi(\mathbf{x}_i) \in \mathcal{H}$ with a finite-dimensional approximation $\widetilde{\phi}_i = \widetilde{\varphi}(\mathbf{x}_i) \in \mathbb{R}^m$.

**Nyström kernel $k$-means** Given a dataset $\mathcal{D}$, we denote with $\mathcal{I} = \{\phi_j\}_{j=1}^m$ a *dictionary* (i.e., subset) of $m$ points $\phi_j$ from $\mathcal{D}$, and with $\Phi_m : \mathbb{R}^m \to \mathcal{H}$ the map with these points as columns. These points acts as landmarks [36], inducing a smaller space $\mathcal{H}_m = \mathrm{Im}(\Phi_m)$ spanned by the dictionary. As we will see in the next section, $\mathcal{I}$ should be chosen so that $\mathcal{H}_m$ is close to the whole span $\mathcal{H}_n = \mathrm{Im}(\Phi_n)$ of the dataset.

Let $\mathbf{K}_{m,m} \in \mathbb{R}^{m \times m}$ be the empirical kernel matrix between all points in $\mathcal{I}$, and denote with

$$\mathbf{\Pi}_m = \Phi_m \Phi_m^\top (\Phi_m \Phi_m^\top)^+ = \Phi_m \mathbf{K}_{m,m}^+ \Phi_m^\top, \tag{1}$$

the orthogonal projection on $\mathcal{H}_m$. Then we can define an approximate ERM over $\mathcal{H}_m$ as

$$\mathbf{C}_{n,m} = \arg\min_{\mathbf{C} \in \mathcal{H}_m^k} \frac{1}{n} \sum_{i=1}^n \min_{j=[k]} \|\phi_i - \mathbf{c}_j\|^2 = \arg\min_{\mathbf{C} \in \mathcal{H}_m^k} \frac{1}{n} \sum_{i=1}^n \min_{j=[k]} \|\mathbf{\Pi}_m(\phi_i - \mathbf{c}_j)\|^2, \tag{2}$$

since any component outside of $\mathcal{H}_m$ is just a constant in the minimization. Note that the centroids $\mathbf{C}_{n,m}$ are still points in $\mathcal{H}_m \subset \mathcal{H}$, and we cannot directly compute them. Instead, we can use the eigen-decomposition of $\mathbf{K}_{m,m} = \mathbf{U}\mathbf{\Lambda}\mathbf{U}^\top$ to rewrite $\mathbf{\Pi}_m = \Phi_m \mathbf{U}\mathbf{\Lambda}^{-1/2}\mathbf{\Lambda}^{-1/2}\mathbf{U}^\top \Phi_m^\top$. Defining now $\widetilde{\varphi}(\cdot) = \mathbf{\Lambda}^{-1/2}\mathbf{U}^\top \Phi_m^\top \varphi(\cdot)$ we have a finite-rank embedding into $\mathbb{R}^m$. Substituting in Eq. (2)

$$\|\mathbf{\Pi}_m(\phi_i - \mathbf{c}_j)\|^2 = \|\mathbf{\Lambda}^{-1/2}\mathbf{U}^\top \Phi_m^\top (\phi_i - \mathbf{c}_j)\|^2 = \|\widetilde{\phi}_i - \mathbf{\Lambda}^{-1/2}\mathbf{U}^\top \Phi_m^\top \mathbf{c}_j\|^2,$$

where $\widetilde{\phi}_i := \mathbf{\Lambda}^{-1/2}\mathbf{U}^\top \Phi_m^\top \phi_i$ are the embedded points. Replacing $\widetilde{\mathbf{c}}_j := \mathbf{\Lambda}^{-1/2}\mathbf{U}^\top \Phi_m^\top \mathbf{c}_j$ and searching over $\widetilde{\mathbf{C}} \in \mathbb{R}^{m \times k}$ instead of searching over $\mathbf{C} \in \mathcal{H}_m^k$, we obtain (similarly to Proposition 2)

$$\widetilde{\mathbf{C}}_{n,m} = \arg\min_{\widetilde{\mathbf{C}} \in \mathbb{R}^{m \times k}} \frac{1}{n} \sum_{i=1}^n \min_{j=[k]} \|\widetilde{\phi}_i - \widetilde{\mathbf{c}}_j\|^2 = \frac{1}{n} \min_{\mathcal{V}} \sum_{j=1}^k \sum_{i \in \mathcal{C}_j} \left\| \widetilde{\phi}_i - \frac{1}{|\mathcal{C}_j|} \sum_{s \in \mathcal{C}_j} \widetilde{\phi}_s \right\|^2, \tag{3}$$

where we do not need to resort to kernel tricks, but can use the $m$-dimensional embeddings $\widetilde{\phi}_i$ to explicitly compute the centroid $\sum_{s \in \mathcal{C}_j} \widetilde{\phi}_s$. Eq. (3) can now be solved in multiple ways. The most straightforward is to run a parametric $k$-means algorithm to compute $\widetilde{\mathbf{C}}_{n,m}$, and then invert the relationship $\widetilde{\mathbf{c}}_j = \Phi_m \mathbf{U}\mathbf{\Lambda}^{-1/2}\mathbf{c}_j$ to bring back the solution to $\mathcal{H}_m$, i.e., $\mathbf{C}_{n,m} = \Phi_m^+ \mathbf{U}^\top \mathbf{\Lambda}^{1/2}\widetilde{\mathbf{C}}_{n,m} = \Phi_m \mathbf{U}\mathbf{\Lambda}^{-1/2}\widetilde{\mathbf{C}}_{n,m}$. This can be done in in $\mathcal{O}(nm)$ space and $\mathcal{O}(nmkt + nm^2)$ time using $t$ steps of Lloyd's algorithm [22] for $k$ clusters. More in detail, computing the embeddings $\widetilde{\phi}_i$ is a one-off cost taking $nm^2$ time. Once the $m$-rank Nyström embeddings $\widetilde{\phi}_i$ are computed they can be stored and manipulated in $nm$ time and space, with an $n/m$ improvement over the $n^2$ time and space required to construct $\mathbf{K}_n$.

## 3.1 Uniform sampling for dictionary construction

Due to its derivation, the computational cost of Algorithm 1 depends on the size $m$ of the dictionary $\mathcal{I}$. Therefore, for computational reasons we would prefer to select as small a dictionary as possible. As a conflicting goal, we also wish to optimize $W(\cdot, \mu_n)$ well, which requires a $\widetilde{\varphi}(\cdot)$ and $\mathcal{I}$ rich enough to approximate $W(\cdot, \mu_n)$ well. Let $\mathbf{\Pi}_m^\perp$ be the projection orthogonal to $\mathcal{H}_m$. Then when $\mathbf{c}_i \in \mathcal{H}_m$

$$\|\phi_i - \mathbf{c}_i\|^2 = \|(\mathbf{\Pi}_m + \mathbf{\Pi}_m^\perp)(\phi_i - \mathbf{c}_i)\|^2 = \|\mathbf{\Pi}_m(\phi_i - \mathbf{c}_i)\|^2 + \|\mathbf{\Pi}_m^\perp \phi_i\|^2.$$

We will now introduce the concept of a $\gamma$-preserving dictionary $\mathcal{I}$ to control the quantity $\|\mathbf{\Pi}_m^\perp \phi_i\|^2$.

---
**Algorithm 1** Nyström Kernel K-Means
---
**Input:** dataset $\mathcal{D} = \{\phi_i\}_{i=1}^n$, dictionary $\mathcal{I} = \{\phi_j\}_{j=1}^m$ with points from $\mathcal{D}$, number of clusters $k$

  compute kernel matrix $\mathbf{K}_{m,m} = \Phi_m^\intercal \Phi_m$ between all points in $\mathcal{I}$

  compute eigenvectors $\mathbf{U}$ and eigenvalues $\mathbf{\Lambda}$ of $\mathbf{K}_{m,m}$

  for each point $\phi_i$, compute embedding $\widetilde{\phi}_i = \mathbf{\Lambda}^{-1/2} \mathbf{U}^\intercal \Phi_m^\intercal \phi_i = \mathbf{\Lambda}^{-1/2} \mathbf{U}^\intercal \mathbf{K}_{m,i} \in \mathbb{R}^m$

  compute optimal centroids $\widetilde{\mathbf{C}}_{n,m} \in \mathbb{R}^{m \times k}$ on the embedded dataset $\widetilde{\mathcal{D}} = \{\widetilde{\phi}_i\}_{i=1}^n$

  compute explicit representation of centroids $\mathbf{C}_{n,m} = \Phi_m \mathbf{U} \mathbf{\Lambda}^{-1/2} \widetilde{\mathbf{C}}_{n,m}$
---

**Definition 2.** *We define the subspace $\mathcal{H}_m$ and dictionary $\mathcal{I}$ as $\gamma$-preserving w.r.t. space $\mathcal{H}_n$ if*

$$\mathbf{\Pi}_m^\perp = \mathbf{\Pi}_n - \mathbf{\Pi}_m \preceq \frac{\gamma}{1-\varepsilon} \left( \Phi_n \Phi_n^\intercal + \gamma \mathbf{\Pi}_n \right)^{-1}. \tag{4}$$

Notice that the inverse $\left( \Phi_n \Phi_n^\intercal + \gamma \mathbf{\Pi}_n \right)^{-1}$ on the right-hand side of the inequality is crucial to control the error $\|\mathbf{\Pi}_m^\perp \phi_i\|^2 \lesssim \gamma \phi_i^\intercal \left( \Phi_n \Phi_n^\intercal + \gamma \mathbf{\Pi}_n \right)^{-1} \phi_i$. In particular, since $\phi_i \in \Phi_n$, we have that in the *worst case* the error is bounded as $\phi_i^\intercal \left( \Phi_n \Phi_n^\intercal + \gamma \mathbf{\Pi}_n \right)^{-1} \phi_i \leq \phi_i^\intercal \left( \phi_i \phi_i^\intercal \right)^+ \phi_i \leq 1$. Conversely, since $\lambda_{\max}(\Phi_n \Phi_n^\intercal) \leq \kappa^2 n$ we know that in the best case the error can be reduced up to $1/n \leq \phi_i^\intercal \phi_i / \lambda_{\max}(\Phi_n \Phi_n^\intercal) \leq \phi_i^\intercal \left( \Phi_n \Phi_n^\intercal + \gamma \mathbf{\Pi}_n \right)^{-1} \phi_i$. Note that the directions associated with the larger eigenvalues are the ones that occur most frequently in the data. As a consequence, Definition 2 guarantees that the overall error across the whole dataset remains small. In particular, we can control the residual $\mathbf{\Pi}_m^\perp \Phi_n$ after the projection as $\|\mathbf{\Pi}_m^\perp \Phi_n\|^2 \leq \gamma \|\Phi_n^\intercal (\Phi_n \Phi_n^\intercal + \gamma \mathbf{\Pi}_n)^{-1} \Phi_n\| \leq \gamma$.

To construct $\gamma$-preserving dictionaries we focus on a uniform random sampling approach[7]. Uniform sampling is historically the first [36], and usually the simplest approach used to construct $\mathcal{I}$. Leveraging results from the literature [7, 14, 25] we can show that uniformly sampling $\widetilde{\mathcal{O}}(n/\gamma)$ landmarks generates a $\gamma$-preserving dictionary with high probability.

**Lemma 1.** *For a given $\gamma$, construct $\mathcal{I}$ by uniformly sampling $m \geq 12\kappa^2 n/\gamma \log(n/\delta)/\varepsilon^2$ landmarks from $\mathcal{D}$. Then w.p. at least $1 - \delta$ the dictionary $\mathcal{I}$ is $\gamma$-preserving.*

Musco and Musco [25] obtains a similar result, but instead of considering the operator $\mathbf{\Pi}_n$ they focus on the finite-dimensional eigenvectors of $\mathbf{K}_n$. Moreover, their $\mathbf{\Pi}_n \preceq \mathbf{\Pi}_m + \frac{\varepsilon \gamma}{1-\varepsilon} \left( \Phi_n \Phi_n^\intercal \right)^+$ bound is weaker and would not be sufficient to satisfy our definition of $\gamma$-accuracy. A result equivalent to Lemma 1 was obtained by Alaoui and Mahoney [3], but they also only focus on the finite-dimensional eigenvectors of $\mathbf{K}_n$, and did not investigate the implications for $\mathcal{H}$.

*Proof sketch of Lemma 1.* It is well known [7, 14] that uniformly sampling $\mathcal{O}(n/\gamma\varepsilon^{-2} \log(n/\delta))$ points with replacement is sufficient to obtain w.p. $1 - \delta$ the following guarantees on $\Phi_m$

$$(1-\varepsilon)\Phi_n \Phi_n^\intercal - \varepsilon\gamma \mathbf{\Pi}_n \preceq \frac{n}{m} \Phi_m \Phi_m^\intercal \preceq (1+\varepsilon)\Phi_n \Phi_n^\intercal + \varepsilon\gamma \mathbf{\Pi}_n.$$

Which implies

$$\left( \frac{n}{m} \Phi_m \Phi_m^\intercal + \gamma \mathbf{\Pi}_n \right)^{-1} \preceq \left( (1-\varepsilon)\Phi_n \Phi_n^\intercal - \varepsilon\gamma \mathbf{\Pi}_n + \gamma \mathbf{\Pi}_n \right)^{-1} = \frac{1}{1-\varepsilon} \left( \Phi_n \Phi_n^\intercal + \gamma \mathbf{\Pi}_n \right)^{-1}$$

We can now rewrite $\mathbf{\Pi}_n$ as

$$\begin{aligned}
\mathbf{\Pi}_n &= \left( \frac{n}{m} \Phi_m \Phi_m^\intercal + \gamma \mathbf{\Pi}_n \right) \left( \frac{n}{m} \Phi_m \Phi_m^\intercal + \gamma \mathbf{\Pi}_n \right)^{-1} \\
&= \frac{n}{m} \Phi_m \Phi_m^\intercal \left( \frac{n}{m} \Phi_m \Phi_m^\intercal + \gamma \mathbf{\Pi}_n \right)^{-1} + \gamma \left( \frac{n}{m} \Phi_m \Phi_m^\intercal + \gamma \mathbf{\Pi}_n \right)^{-1} \\
&\preceq \frac{n}{m} \Phi_m \Phi_m^\intercal \left( \frac{n}{m} \Phi_m \Phi_m^\intercal \right)^+ + \gamma \left( \frac{n}{m} \Phi_m \Phi_m^\intercal + \gamma \mathbf{\Pi}_n \right)^{-1} \\
&= \mathbf{\Pi}_m + \gamma \left( \frac{n}{m} \Phi_m \Phi_m^\intercal + \gamma \mathbf{\Pi}_n \right)^{-1} \preceq \mathbf{\Pi}_m + \frac{\gamma}{1-\varepsilon} \left( \Phi_n \Phi_n^\intercal + \gamma \mathbf{\Pi}_n \right)^{-1} \qquad \square
\end{aligned}$$

In other words, using uniform sampling we can reduce the size of the search space $\mathcal{H}_m$ by a $1/\gamma$ factor (from $n$ to $m \simeq n/\gamma$) in exchange for a $\gamma$ additive error, resulting in a computation/approximation trade-off that is linear in $\gamma$.

## 4  Theoretical analysis

Exploiting the error bound for $\gamma$-preserving dictionaries we are now ready for the main result of this paper: showing that we can improve the computational aspect of kernel $k$-means using Nyström embedding, while maintaining optimal generalization guarantees.

**Theorem 1.** *Given a $\gamma$-preserving dictionary*

$$\mathcal{E}(\mathbf{C}_{n,m}) = W(\mathbf{C}_{n,m}, \mu) - W(\mathbf{C}_n, \mu) \leq \mathcal{O}\left(k\left(\frac{1}{\sqrt{n}} + \frac{\gamma}{n}\right)\right)$$

From a statistical point of view, Theorem 1 shows that if $\mathcal{I}$ is $\gamma$-preserving, the ERM in $\mathcal{H}_m$ achieves the same *excess* risk as the exact ERM from $\mathcal{H}_n$ up to an additional $\gamma/n$ error. Therefore, choosing $\gamma = \sqrt{n}$ the solution $\mathbf{C}_{n,m}$ achieves the $\mathcal{O}(k/\sqrt{n}) + \mathcal{O}(k\sqrt{n}/n) \leq \mathcal{O}(k/\sqrt{n})$ generalization [10].

From a computational point of view, Lemma 1 shows that we can construct an $\sqrt{n}$-preserving dictionary simply by sampling $\widetilde{\mathcal{O}}(\sqrt{n})$ points uniformly[1], which greatly reduces the embedding size from $n$ to $\sqrt{n}$, and the total required space from $n^2$ to $\widetilde{\mathcal{O}}(n\sqrt{n})$.

Time-wise, the bottleneck becomes the construction of the embeddings $\widetilde{\phi}_i$, which takes $nm^2 \leq \widetilde{\mathcal{O}}(n^2)$ time, while each iterations of Lloyd's algorithm only requires $nm \leq \widetilde{\mathcal{O}}(n\sqrt{n})$ time. In the full generality of our setting this is practically optimal, since computing a $\sqrt{n}$-preserving dictionary is in general as hard as matrix multiplication [26, 9], which requires $\Omega(n^2)$ time. In other words, unlike the case of space complexity, there is no free lunch for time complexity, that in the worst case must scale as $n^2$ similarly to the exact case. Nonetheless embedding the points is an embarrassingly parallel problem that can be easily distributed, while in practice it is usually the execution of the Lloyd's algorithm that dominates the runtime.

Finally, when the dataset satisfies certain regularity conditions, the size of $\mathcal{I}$ can be improved, which reduces both embedding and clustering runtime. Denote with $d_{\text{eff}}^n(\gamma) = \text{Tr}\left(\mathbf{K}_n^{\mathsf{T}}(\mathbf{K}_n + \mathbf{I}_n)^{-1}\right)$ the so-called *effective* dimension [3] of $\mathbf{K}_n$. Since $\text{Tr}\left(\mathbf{K}_n^{\mathsf{T}}(\mathbf{K}_n + \mathbf{I}_n)^{-1}\right) \leq \text{Tr}\left(\mathbf{K}_n^{\mathsf{T}}(\mathbf{K}_n)^+\right)$, we have that $d_{\text{eff}}^n(\gamma) \leq r := \text{Rank}(\mathbf{K}_n)$, and therefore $d_{\text{eff}}^n(\gamma)$ can be seen as a soft version of the rank. When $d_{\text{eff}}^n(\gamma) \ll \sqrt{n}$ it is possible to construct a $\gamma$-preserving dictionary with only $d_{\text{eff}}^n(\gamma)$ landmarks in $\widetilde{\mathcal{O}}(nd_{\text{eff}}^n(\gamma)^2)$ time using specialized algorithms [14] (see Section 6). In this case, the embedding step would require only $\widetilde{\mathcal{O}}(nd_{\text{eff}}^n(\gamma)^2) \ll \widetilde{\mathcal{O}}(n^2)$, improving both time and space complexity.

Morever, to the best of our knowledge, this is the first example of an unsupervised non-parametric problem where it is always (i.e., without assumptions on $\mu$) possible to preserve the optimal $\mathcal{O}(1/\sqrt{n})$ risk rate while reducing the search from the whole space $\mathcal{H}$ to a smaller $\mathcal{H}_m$ subspace.

*Proof sketch of Theorem 1.* We can separate the distance between $W(\mathbf{C}_{n,m}, \mu) - W(\mathbf{C}_n, \mu)$ in a component that depends on how close $\mu$ is to $\mu_n$, bounded using Proposition 1, and a component $W(\mathbf{C}_{n,m}, \mu_n) - W(\mathbf{C}_n, \mu_n)$ that depends on the distance between $\mathcal{H}_n$ and $\mathcal{H}_m$

**Lemma 2.** *Given a $\gamma$-preserving dictionary*

$$W(\mathbf{C}_{n,m}, \mu_n) - W(\mathbf{C}_n, \mu_n) \leq \frac{\min(k, d_{\text{eff}}^n(\gamma))}{1 - \varepsilon}\frac{\gamma}{n}$$

To show this we can rewrite the objective as (see [17])

$$W(\mathbf{C}_{n,m}, \mu_n) = \|\Phi_n - \mathbf{\Pi}_m \Phi_n \mathbf{S}_{n,m}\|_F^2 = \text{Tr}(\Phi_n^{\mathsf{T}}\Phi_n - \mathbf{S}_n \Phi_n^{\mathsf{T}} \mathbf{\Pi}_m \Phi_n \mathbf{S}_n),$$

where $\mathbf{S}_n \in \mathbb{R}^{n \times n}$ is a $k$-rank *projection* matrix associated with the *exact* clustering $\mathbf{C}_n$. Then using Definition 2 we have $\mathbf{\Pi}_m - \mathbf{\Pi}_n \succeq -\frac{\gamma}{1-\varepsilon}(\Phi_n\Phi_n^{\mathsf{T}} + \gamma\mathbf{\Pi}_n)^{-1}$ and we obtain an *additive* error bound

$$\text{Tr}(\Phi_n^{\mathsf{T}}\Phi_n - \mathbf{S}_n\Phi_n^{\mathsf{T}}\mathbf{\Pi}_m\Phi_n\mathbf{S}_n)$$
$$\leq \text{Tr}\left(\Phi_n^{\mathsf{T}}\Phi_n - \mathbf{S}_n\Phi_n^{\mathsf{T}}\Phi_n\mathbf{S}_n + \frac{\gamma}{1-\varepsilon}\mathbf{S}_n\Phi_n^{\mathsf{T}}(\Phi_n\Phi_n^{\mathsf{T}} + \gamma\mathbf{\Pi}_n)^{-1}\Phi_n\mathbf{S}_n\right)$$
$$= W(\mathbf{C}_n, \mu_n) + \frac{\gamma}{1-\varepsilon}\text{Tr}\left(\mathbf{S}_n\Phi_n^{\mathsf{T}}(\Phi_n\Phi_n^{\mathsf{T}} + \gamma\mathbf{\Pi}_n)^{-1}\Phi_n\mathbf{S}_n\right).$$

Since $\|\Phi_n^\mathsf{T}(\Phi_n\Phi_n^\mathsf{T} + \gamma\mathbf{\Pi}_n)^{-1}\Phi_n\| \le 1$, $\mathbf{S}_n$ is a projection matrix, and $\mathrm{Tr}(\mathbf{S}_n) = k$ we have

$$\tfrac{\gamma}{1-\varepsilon}\,\mathrm{Tr}\left(\mathbf{S}_n\Phi_n^\mathsf{T}(\Phi_n\Phi_n^\mathsf{T} + \gamma\mathbf{\Pi}_n)^{-1}\Phi_n\mathbf{S}_n\right) \le \tfrac{\gamma}{1-\varepsilon}\,\mathrm{Tr}\left(\mathbf{S}_n\mathbf{S}_n\right) = \tfrac{\gamma k}{1-\varepsilon}.$$

Conversely, if we focus on the matrix $\Phi_n^\mathsf{T}(\Phi_n\Phi_n^\mathsf{T} + \gamma\mathbf{\Pi}_n)^{-1}\Phi_n \preceq \mathbf{\Pi}_n$ we have

$$\tfrac{\gamma}{1-\varepsilon}\,\mathrm{Tr}\left(\mathbf{S}_n\Phi_n^\mathsf{T}(\Phi_n\Phi_n^\mathsf{T} + \mathbf{\Pi}_n)^{-1}\Phi_n\mathbf{S}_n\right) \le \tfrac{\gamma}{1-\varepsilon}\,\mathrm{Tr}\left(\Phi_n^\mathsf{T}(\Phi_n\Phi_n^\mathsf{T} + \mathbf{\Pi}_n)^{-1}\Phi_n\right) \le \tfrac{\gamma d_{\mathrm{eff}}^n(\gamma)}{1-\varepsilon}.$$

Since both bounds hold simultaneously, we can simply take the minimum to conclude our proof. $\qquad\square$

We now compare the theorem with previous work. Many approximate kernel $k$-means methods have been proposed over the years, and can be roughly split in two groups.

Low-rank *decomposition* based methods try to directly simplify the optimization problem from Proposition 2, replacing the kernel matrix $\mathbf{K}_n$ with an approximate $\widetilde{\mathbf{K}}_n$ that can be stored and manipulated more efficiently. Among these methods we can mention partial decompositions [8], Nyström approximations based on uniform [36], $k$-means++ [27], or ridge leverage score (RLS) sampling[35, 25, 14], and random-feature approximations [6]. None of these optimization based methods focus on the underlying excess risk problem, and their analysis cannot be easily integrated in existing results, as the approximate minimum found has no clear interpretation as a statistical ERM.

Other works take the same *embedding* approach that we do, and directly replace the exact $\varphi(\cdot)$ with an approximate $\widetilde{\varphi}(\cdot)$, such as Nyström embeddings [36], Gaussian projections [10], and again random-feature approximations [29]. Note that these approaches also result in approximate $\widetilde{\mathbf{K}}_n$ that can be manipulated efficiently, but are simpler to analyze theoretically. Unfortunately, no existing embedding based methods can guarantee at the same time optimal excess risk rates and a reduction in the size of $\mathcal{H}_m$, and therefore a reduction in computational cost.

To the best of our knowledge, the only other result providing excess risk guarantee for approximate kernel $k$-means is Biau et al. [10], where the authors consider the excess risk of the ERM when the approximate $\mathcal{H}_m$ is obtained using Gaussian projections. Biau et al. [10] notes that the feature map $\varphi(\mathbf{x}) = \sum_{s=1}^{D}\psi_s(\mathbf{x})$ can be expressed using an expansion of basis functions $\psi_s(\mathbf{x})$, with $D$ very large or infinite. Given a matrix $\mathbf{P} \in \mathbb{R}^{m\times D}$ where each entry is a standard Gaussian r.v., [10] proposes the following $m$-dimensional approximate feature map $\widetilde{\varphi}(\mathbf{x}) = \mathbf{P}[\psi_1(\mathbf{x}),\ldots,\psi_D(\mathbf{x})] \in \mathbb{R}^m$. Using Johnson-Lindenstrauss (JL) lemma [19], they show that if $m \ge \log(n)/\nu^2$ then a multiplicative error bound of the form $W(\mathbf{C}_{n,m},\mu_n) \le (1+\nu)W(\mathbf{C}_n,\mu_n)$ holds. Reformulating their bound, we obtain that $W(\mathbf{C}_{n,m},\mu_n) - W(\mathbf{C}_n,\mu_n) \le \nu W(\mathbf{C}_n,\mu_n) \le \nu\kappa^2$ and $\mathcal{E}(\mathbf{C}_{n,m}) \le \mathcal{O}(k/\sqrt{n} + \nu)$.

Note that to obtain a bound comparable to Theorem 1, and if we treat $k$ as a constant, we need to take $\nu = \gamma/n$ which results in $m \ge (n/\gamma)^2$. This is always worse than our $\widetilde{\mathcal{O}}(n/\gamma)$ result for uniform Nyström embedding. In particular, in the $1/\sqrt{n}$ risk rate setting Gaussian projections would require $\nu = 1/\sqrt{n}$ resulting in $m \ge n\log(n)$ random features, which would not bring any improvement over computing $\mathbf{K}_n$. Moreover when $D$ is infinite, as it is usually the case in the non-parametric setting, the JL projection is not explicitly computable in general and Biau et al. [10] must assume the existence of a computational oracle capable of constructing $\widetilde{\varphi}(\cdot)$. Finally note that, under the hood, traditional embedding methods such as those based on JL lemma, usually provide only bounds of the form $\mathbf{\Pi}_n - \mathbf{\Pi}_m \preceq \gamma\mathbf{\Pi}_n$, and an error $\|\mathbf{\Pi}_m^\perp\phi_i\|^2 \le \gamma\|\phi_i\|^2$ (see the discussion of Definition 2). Therefore the error can be larger along multiple directions, and the overall error $\|\mathbf{\Pi}_m^\perp\Phi_n\|^2$ across the dictionary can be as large as $n\gamma$ rather than $\gamma$.

Recent work in RLS sampling has also focused on bounding the distance $W(\mathbf{C}_{n,m},\mu_n) - W(\mathbf{C}_n,\mu_n)$ between empirical errors. Wang et al. [35] and Musco and Musco [25] provide multiplicative error bounds of the form $W(\mathbf{C}_{n,m},\mu_n) \le (1+\nu)W(\mathbf{C}_n,\mu_n)$ for uniform and RLS sampling. Nonetheless, they only focus on empirical risk and do not investigate the interaction between approximation and generalization, i.e., statistics and computations. Moreover, as we already remarked for [10], to achieve the $1/\sqrt{n}$ excess risk rate using a multiplicative error bound we would require an unreasonably small $\nu$, resulting in a large $m$ that brings no computational improvement over the exact solution.

Finally, note that when [31] showed that a favourable trade-off was possible for kernel ridge regression (KRR), they strongly leveraged the fact that KRR is a $\gamma$-regularized problem. Therefore, all eigenvalues and eigenvectors in the $\Phi_n\Phi_n^\mathsf{T}$ covariance matrix smaller than the $\gamma$ regularization do not influence significantly the solution. Here we show the same for kernel $k$-means, a problem

without regularization. This hints at a deeper geometric motivation which might be at the root of both problems, and potentially similar approaches could be leveraged in other domains.

## 4.1 Further results: beyond ERM

So far we provided guarantees for $\mathbf{C}_{n,m}$, that this the ERM in $\mathcal{H}_m$. Although $\mathcal{H}_m$ is much smaller than $\mathcal{H}_n$, solving the optimization problem to find the ERM is still NP-Hard in general [4]. Nonetheless, Lloyd's algorithm [22], when coupled with a careful $k$-means++ seeding, can return a good approximate solution $\mathbf{C}_{n,m}^{++}$.

**Proposition 3** ([5]). *For any dataset* $\mathbb{E}_{\mathcal{A}}[W(\mathbf{C}_{n,m}^{++}, \mu_n)] \leq 8(\log(k) + 2)W(\mathbf{C}_{n,m}, \mu_n)$, *where* $\mathcal{A}$ *is the randomness deriving from the k-means++ initialization.*

Note that, similarly to [35, 25], this is a multiplicative error bound on the empirical risk, and as we discussed we cannot leverage Lemma 2 to bound the excess risk $\mathcal{E}(\mathbf{C}_{n,m}^{++})$. Nonetheless we can still leverage Lemma 2 to bound only the expected risk $W(\mathbf{C}_{n,m}^{++}, \mu)$, albeit with an extra error term appearing that scales with the optimal clustering risk $W^*(\mu)$ (see Proposition 1).

**Theorem 2.** *Given a* $\gamma$-*preserving dictionary*

$$\mathbb{E}_{\mathcal{D} \sim \mu} \left[ \mathbb{E}_{\mathcal{A}}[W(\mathbf{C}_{n,m}^{++}, \mu)] \right] \leq \mathcal{O} \left( \log(k) \left( \frac{k}{\sqrt{n}} + k\frac{\gamma}{n} + W^*(\mu) \right) \right).$$

From a statistical perspective, we can once again, set $\gamma = \sqrt{n}$ to obtain a $\mathcal{O}(k/\sqrt{n})$ rate for the first part of the bound. Conversely, the optimal clustering risk $W^*(\mu)$ is a $\mu$-dependent quantity that cannot in general be bounded in $n$, and captures how well our model, i.e., the choice of $\mathcal{H}$ and how well the criterion $W(\cdot, \mu)$, matches reality.

From a computational perspective, we can now bound the computational cost of finding $\mathbf{C}_{n,m}^{++}$. In particular, each iteration of Lloyd's algorithm will take only $\widetilde{\mathcal{O}}(n\sqrt{n}k)$ time. Moreover, when $k$-means++ initialization is used, the expected number of iterations required for Lloyd's algorithm to converge is only logarithmic [1]. Therefore, ignoring the time required to embed the points, we can find a solution in $\widetilde{\mathcal{O}}(n\sqrt{n}k)$ time and space instead of the $\widetilde{\mathcal{O}}(n^2k)$ cost required by the exact method, with a strong $\mathcal{O}(\sqrt{n})$ improvement.

Finally, if the data distribution satisfies some regularity assumption the following result follows [15].

**Corollary 1.** *If we denote by* $\mathcal{X}_\mu$ *the support of the distribution* $\mu$ *and assume* $\varphi(\mathcal{X}_\mu)$ *to be a* $d$-*dimensional manifold, then* $W^*(\mu) \leq dk^{-2/d}$, *and given a* $\sqrt{n}$-*preserving dictionary the expected cost satisfies* $\mathbb{E}_{\mathcal{D} \sim \mu}[\mathbb{E}_{\mathcal{A}}[W(\mathbf{C}_{n,m}^{++}, \mu)]] \leq \mathcal{O}\left( \log(k) \left( \frac{k}{\sqrt{n}} + dk^{-2/d} \right) \right)$.

## 5 Experiments

We now evaluate experimentally the claims of Theorem 1, namely that sampling $\widetilde{\mathcal{O}}(n/\gamma)$ increases the excess risk by an extra $\gamma/n$ factor, and that $m = \sqrt{n}$ is sufficient to recover the optimal rate. We use the `Nystroem` and `MiniBatchKmeans` classes from the `sklearn` python library [28]to implement kernel $k$-means with Nyström embedding (Algorithm 1) and we compute the solution $\mathbf{C}_{n,m}^{++}$.

For our experiments we follow the same approach as Wang et al. [35], and test our algorithm on two variants of the MNIST digit dataset. In particular, MNIST60K [20] is the original MNIST dataset containing pictures each with $d = 784$ pixels. We divide each pixel by 255, bringing each feature in a $[0, 1]$ interval. We split the dataset in two part, $n = 60000$ samples are used to compute the $W(\mathbf{C}_{n,m}^{++})$ centroids, and we leave out unseen 10000 samples to compute $W(\mathbf{C}_{n,m}^{++}, \mu_{test})$, as a proxy for $W(\mathbf{C}_{n,m}^{++}, \mu)$. To test the scalability of our approach we also consider the MNIST8M dataset from the infinite MNIST project [23], constructed using non-trivial transformations and corruptions of the original MNIST60K images. Here we compute $\mathbf{C}_{n,m}^{++}$ using $n = 8000000$ images, and compute $W(\mathbf{C}_{n,m}^{++}, \mu_{test})$ on 100000 unseen images. As in Wang et al. [35] we use Gaussian kernel with bandwidth $\sigma = (1/n^2)\sqrt{\sum_{i,j} \|\mathbf{x}_i - \mathbf{x}_j\|^2}$.

**MNIST60K:** these experiments are small enough to run in less than a minute on a single laptop with 4 cores and 8GB of RAM. The results are reported in Fig. 1. On the left we report in blue

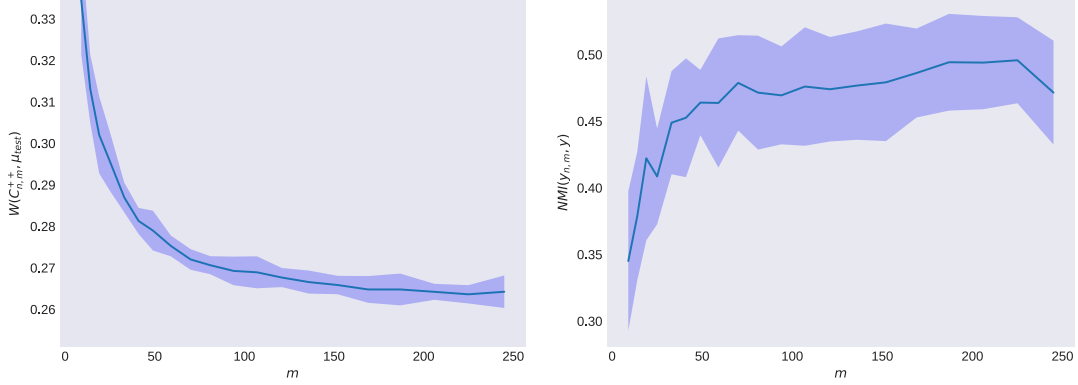

Figure 1: Results for MNIST60K

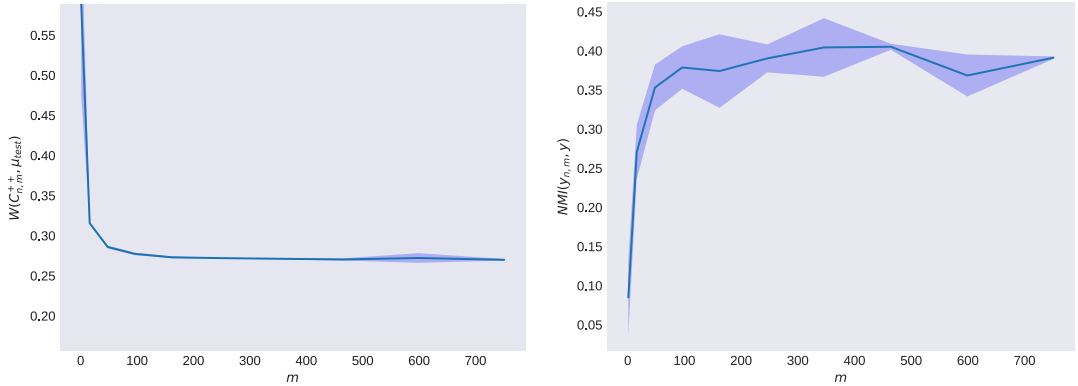

Figure 2: Results for MNIST8M

$W(\mathbf{C}_{n,m}^{++}, \mu_{test})$, where the shaded region is a 95% confidence interval for the mean over 10 runs. As predicted, the expected cost decreases as the size of $\mathcal{H}_m$ increases, and plateaus once we achieve $1/m \simeq 1/\sqrt{n}$, in line with the statistical error. Note that the normalized mutual information (NMI) between the true $[0 - 9]$ digit classes $\mathbf{y}$ and the computed cluster assignments $\mathbf{y}_{n,m}$ also plateaus around $1/\sqrt{n}$. While this is not predicted by the theory, it strengthens the intuition that beyond a certain capacity expanding $\mathcal{H}_m$ is computationally wasteful.

**MNIST8M:** to test the scalability of our approach, we run the same experiment on millions of points. Note that we carry out our MNIST8M experiment on a *single* 36 core machine with 128GB of RAM, much less than the setup of [35], where at minimum a cluster of 8 such nodes are used. The behaviour of $W(\mathbf{C}_{n,m}^{++}, \mu_{test})$ and NMI are similar to MNIST60K, with the increase in dataset size allowing for stronger concentration and smaller confidence intervals. Finalle, note that around $m = 400$ uniformly sampled landmarks are sufficient to achieve $NMI(\mathbf{y}_{n,m}, \mathbf{y}) = 0.405$, matching the $0.406$ NMI reported by [35] for a larger $m = 1600$, although smaller than the $0.423$ NMI they report for $m = 1600$ when using a slower, PCA based method to compute the embeddings, and RLS sampling to select the landmarks. Nonetheless, computing $\mathbf{C}_{n,m}^{++}$ takes less than 6 minutes on a single machine, while their best solution required more than 1.5hr on a cluster of 32 machines.

## 6  Open questions and conclusions

Combining Lemma 1 and Lemma 2, we know that using uniform sampling we can linearly trade-off a $1/\gamma$ decrease in sub-space size $m$ with a $\gamma/n$ increase in excess risk. While this is sufficient to maintain the $\mathcal{O}(1/\sqrt{n})$ rate, it is easy to see that the same would not hold for a $\mathcal{O}(1/n)$ rate, since we would need to uniformly sample $n/1$ landmarks losing all computational improvements.

To achieve a better trade-off we must go beyond uniform sampling and use different probabilities for each sample, to capture their uniqueness and contribution to the approximation error.

**Definition 3** ([3]). *The $\gamma$-ridge leverage score (RLS) of point $i \in [n]$ is defined as*

$$\tau_i(\gamma) = \boldsymbol{\phi}_i^\mathsf{T}(\boldsymbol{\Phi}_n\boldsymbol{\Phi}_n^\mathsf{T} + \gamma\boldsymbol{\Pi}_n)^{-1}\boldsymbol{\phi}_i = \mathbf{e}_i^\mathsf{T}\mathbf{K}_n(\mathbf{K}_n + \gamma\mathbf{I}_n)^{-1}\mathbf{e}_i. \tag{5}$$

*The sum of the RLSs $d_{\text{eff}}^n(\gamma) = \sum_{i=1}^n \tau_i(\gamma)$ is the empirical effective dimension of the dataset.*

Ridge leverage scores are closely connected to the residual $\|\boldsymbol{\Pi}_m^\perp\boldsymbol{\phi}_i\|^2$ after the projection $\boldsymbol{\Pi}_m$ discussed in Definition 2. In particular, using Lemma 2 we have that the residual can be bounded as $\|\boldsymbol{\Pi}_m^\perp\boldsymbol{\phi}_i\|^2 \leq \frac{\gamma}{1-\varepsilon}\boldsymbol{\phi}_i^\mathsf{T}(\boldsymbol{\Phi}_n\boldsymbol{\Phi}_n^\mathsf{T} + \gamma\boldsymbol{\Pi}_n)^{-1}\boldsymbol{\phi}_i$. It is easy to see that, up to a factor $\frac{\gamma}{1-\varepsilon}$, high-RLS points are also high-residual points. Therefore it is not surprising that sampling according to RLSs quickly selects any high-residual points and covers $\mathcal{H}_n$, generating a $\gamma$-preserving dictionary.

**Lemma 3.** *[11] For a given $\gamma$, construct $\mathcal{I}$ by sampling $m \geq 12\kappa^2 d_{\text{eff}}^n(\gamma)\log(n/\delta)/\varepsilon^2$ landmarks from $\mathcal{D}$ proportionally to their RLS. Then w.p. at least $1 - \delta$ the dictionary $\mathcal{I}$ is $\gamma$-preserving.*

Note there exist datasets where the RLSs are uniform, and therefore in the worst case the two sampling approaches coincide. Nonetheless, when the data is more structured $m \simeq d_{\text{eff}}^n(\gamma)$ can be much smaller than the $n/\gamma$ dictionary size required by uniform sampling.

Finally, note that computing RLSs exactly also requires constructing $\mathbf{K}_n$ and $\mathcal{O}(n^2)$ time and space, but in recent years a number of fast approximate RLSs sampling methods [14] have emerged that can construct $\gamma$-preserving dictionaries of size $\widetilde{\mathcal{O}}(d_{\text{eff}}^n(\gamma))$ in just $\widetilde{\mathcal{O}}(nd_{\text{eff}}^n(\gamma)^2)$ time. Using this result, it is trivial to sharpen the computational aspects of Theorem 1 in special cases.

In particular, we can generate a $\sqrt{n}$-preserving dictionary with only $d_{\text{eff}}^n(\sqrt{n})$ elements instead of the $\sqrt{n}$ required by uniform sampling. Using concentration arguments [31] we also know that w.h.p. the empirical effective dimension is at most three times $d_{\text{eff}}^n(\gamma) \leq 3d_{\text{eff}}^\mu(\gamma)$ the expected effective dimension, a $\mu$-dependent quantity that captures the interaction between $\mu$ and the RKHS $\mathcal{H}$.

**Definition 4.** *Given the expected covariance operator $\boldsymbol{\Psi} := \mathbb{E}_{\mathbf{x}\sim\mu}\left[\boldsymbol{\phi}(\mathbf{x})\boldsymbol{\phi}(\mathbf{x})^\mathsf{T}\right]$, the expected effective dimension is defined as $d_{\text{eff}}^\mu(\gamma) = \mathbb{E}_{\mathbf{x}\sim\mu}\left[\boldsymbol{\phi}(\mathbf{x})\left(\boldsymbol{\Psi} + \gamma\boldsymbol{\Pi}\right)^{-1}\boldsymbol{\phi}(\mathbf{x})\right]$. Moreover, for some constant $c$ that depends only on $\varphi(\cdot)$ and $\mu$, $d_{\text{eff}}^\mu(\gamma) \leq c\left(n/\gamma\right)^\eta$ with $0 < \eta \leq 1$.*

Note that $\eta = 1$ just gives us the $d_{\text{eff}}^\mu(\gamma) \leq \mathcal{O}(n/\gamma)$ worst-case upper bound that we saw for $d_{\text{eff}}^n(\gamma)$, and it is always satisfied when the kernel function is bounded. If instead we have a faster spectral decay, $\eta$ can be much smaller. For example, if the eigenvalues of $\boldsymbol{\Psi}$ decay polynomially as $\lambda_i = i^{-\eta}$, then $d_{\text{eff}}^\mu(\gamma) \leq c\left(n/\gamma\right)^\eta$, and in our case $\gamma = \sqrt{n}$ we have $d_{\text{eff}}^\mu(\sqrt{n}) \leq cn^{\eta/2}$.

We can now better characterize the gap between statistics and computation: using RLSs sampling we can improve the computational aspect of Theorem 1 from $\sqrt{n}$ to $d_{\text{eff}}^\mu(\gamma)$, but the risk rate remains $\mathcal{O}(k/\sqrt{n})$ due to the $\mathcal{O}(k/\sqrt{n})$ component coming from Proposition 1.

Assume for a second we could generalize, with additional assumptions, Proposition 1 to a faster $\mathcal{O}(1/n)$ rate. Then applying Lemma 2 with $\gamma = 1$ we would obtain a risk $\mathcal{E}(\mathbf{C}_{n,m}) \leq \mathcal{O}(k/n) + \mathcal{O}(k/n)$. Here we see how the regularity condition on $d_{\text{eff}}^\mu(1)$ becomes crucial. In particular, if $\eta = 1$, then we have $d_{\text{eff}}^\mu(1) \sim n$ and no gain. If instead $\eta < 1$ we obtain $d_{\text{eff}}^\mu(1) \leq n^\eta$. This kind of adaptive rates were shown to be possible in supervised learning [31], but seems to still be out of reach for approximate kernel $k$-means.

One possible approach to fill this gap is to look at fast $\mathcal{O}(1/n)$ excess risk rates for kernel $k$-means.

**Proposition 4** ([21], informal). *Assume that $k \geq 2$, and that $\mu$ satisfies a margin condition with radius $r_0$. If $\mathbf{C}_n$ is an empirical risk minimizer, then, with probability larger than $1 - e^{-\delta}$,*

$$\mathcal{E}(\mathbf{C}_n) \leq \widetilde{\mathcal{O}}\left(\frac{1}{r_0}\frac{(k + log(|M|))\log(1/\delta)}{n}\right),$$

*where $|M|$ is the cardinality of the set of all optimal (up to a relabeling) clustering.*

For more details on the margin assumption, we refer the reader to the original paper [21]. Intuitively the margin condition asks that every labeling (Voronoi grouping) associated with an optimal clustering is reflected by large separation in $\mu$. This margin condition also acts as a counterpart of the usual margin conditions for supervised learning where $\mu$ must have lower density around the neighborhood of the critical area $\{\mathbf{x}|\mu'(Y = 1|X = \mathbf{x}) = 1/2\}$. Unfortunately, it is not easy to integrate Proposition 4 in our analysis, as it is not clear how the margin condition translate from $\mathcal{H}_n$ to $\mathcal{H}_m$.

**Acknowledgments.**

This material is based upon work supported by the Center for Brains, Minds and Machines (CBMM), funded by NSF STC award CCF-1231216, and the Italian Institute of Technology. We gratefully acknowledge the support of NVIDIA Corporation for the donation of the Titan Xp GPUs and the Tesla k40 GPU used for this research. L. R. acknowledges the support of the AFOSR projects FA9550-17-1-0390 and BAA-AFRL-AFOSR-2016-0007 (European Office of Aerospace Research and Development), and the EU H2020-MSCA-RISE project NoMADS - DLV-777826. A. R. acknowledges the support of the European Research Council (grant SEQUOIA 724063).

## Footnotes

[1] $\widetilde{\mathcal{O}}$ hides logarithmic dependencies on $n$ and $m$.

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
