[Supplementary Material · nips_2018_nystro_kmeans_cr_supplementary.pdf]

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

# A  Proofs

*Proof of Theorem 1.*  Given our dictionary $\mathcal{I}$, we decompose

$$\underset{\mathcal{D}\sim\mu}{\mathbb{E}}[W(\mathbf{C}_{n,m},\mu)] - W^*(\mu) = \underset{\mathcal{D}\sim\mu}{\mathbb{E}}[W(\mathbf{C}_{n,m},\mu) - W(\mathbf{C}_n,\mu)] + \underset{\mathcal{D}\sim\mu}{\mathbb{E}}[W(\mathbf{C}_n,\mu)] - W^*(\mu)$$

We can use Proposition 1 to bound the second pair as $\mathcal{O}(k/\sqrt{n})$ using the proposition. Then we further split

$$
\begin{aligned}
W(\mathbf{C}_{n,m},\mu) - W(\mathbf{C}_n,\mu) &= W(\mathbf{C}_{n,m},\mu) - W(\mathbf{C}_{n,m},\mu_n) & (a)\\
&+ W(\mathbf{C}_{n,m},\mu_n) - W(\mathbf{C}_n,\mu_n) & (b)\\
&+ W(\mathbf{C}_n,\mu_n) - W(\mathbf{C}_n,\mu) & (c).
\end{aligned}
$$

The last line $(c)$ is negative, as $\mathbf{C}_n$ is optimal w.r.t. $W(\cdot,\mu_n)$. The first line $(a)$ can also be bounded by [10, Lemma 4.3], a stronger version of Proposition 1. To bound the middle term $(b)$ we further expand

$$W(\mathbf{C}_{n,m},\mu_n) = \frac{1}{n}\sum_{i=1}^{n}\min_{j=1,\ldots,k}\|\boldsymbol{\phi}_i - \mathbf{c}_{n,m,j}\|^2.$$

Note that $W$ evaluates the minimum $\min_{j=1,\ldots,k}\|\boldsymbol{\phi}_i - \mathbf{c}_{n,m,j}\|^2$, while $\mathbf{C}_{n,m}$ has been constructed to minimize $\min_{j=1,\ldots,k}\|\boldsymbol{\Pi}_m\boldsymbol{\phi}_i - \mathbf{c}_{n,m,j}\|^2$. Nonetheless, since $\mathbf{c}_{n,m,j}\in\mathcal{H}_m$ is orthogonal to $\boldsymbol{\Pi}_m^\perp$

$$
\begin{aligned}
\min_{j=1,\ldots,k}\|\boldsymbol{\phi}_i - \mathbf{c}_{n,m,j}\|^2 &= \min_{j=1,\ldots,k}\|\boldsymbol{\Pi}_m\boldsymbol{\phi}_i + \boldsymbol{\Pi}_m^\perp\boldsymbol{\phi}_i - \mathbf{c}_{n,m,j}\|^2\\
&= \min_{j=1,\ldots,k}\|\boldsymbol{\Pi}_m\boldsymbol{\phi}_i - \mathbf{c}_{n,m,j}\|^2 + \|\boldsymbol{\Pi}_m^\perp\boldsymbol{\phi}_i\|^2 + 2\underbrace{(\boldsymbol{\Pi}_m\boldsymbol{\phi}_i - \mathbf{c}_{n,m,j})^\top\boldsymbol{\Pi}_m^\perp\boldsymbol{\phi}_i}_{0}\\
&= \min_{j=1,\ldots,k}\|\boldsymbol{\Pi}_m\boldsymbol{\phi}_i - \mathbf{c}_{n,m,j}\|^2 + \|\boldsymbol{\Pi}_m^\perp\boldsymbol{\phi}_i\|^2,
\end{aligned}
$$

and both criteria are minimized by the same $j$ (i.e., they assign the point to the same cluster).

Before continuing, we must introduce additional notation to represent $\mathbf{C}_n$ as the average of the points in each cluster. Let $\{\mathbf{f}_j\}_{j=1}^k$ be the cluster indicator vectors such that $[\mathbf{f}_j]_i = 1/|\mathcal{C}_j|$ if sample $i$ is in the $j$-th cluster and $[\mathbf{f}_j]_i = 0$ otherwise. As a consequence $\|\mathbf{f}_j\|^2 = 1/|\mathcal{C}_j|$, and $[\mathbf{f}_j]_i = 1/\|\mathbf{f}_c\|^2$. Denote with $\mathbf{F}\in\mathbb{R}^{n\times k}$ the matrix containing $\mathbf{f}_i$ as columns, and let $\mathcal{S}$ be the space of feasible clustering, such that all $\mathbf{f}_j$ are $\{0, 1/\|\mathbf{f}_c\|^2\}$ binary, and each row of $\mathbf{F}$ contains only one non-zero entry. Let $\mathbf{R}\in\mathbb{R}^{k\times k}$ be the diagonal matrix with $1/\|\mathbf{f}_c\|^2$ as the diagonal entries.

Then $\mathbf{F}^\top\mathbf{F} = \mathbf{R}$ and $\mathbf{S} := \mathbf{F}\mathbf{R}^{-1}\mathbf{F}^\top\in\mathbb{R}^{n\times n}$ is a projection matrix. Finally we can express the centroids as $\mathbf{c}_j = \Phi_n\mathbf{f}_j = (1/|\mathcal{C}_j|)\sum\boldsymbol{\phi}_i$ and $\mathbf{C}_n = \Phi_n\mathbf{F}_n$.

Similarly, we define the optimal $\mathbf{F}_{n,m}$ associated with $\mathbf{C}_{n,m}$, of $\mathbf{R}_{n,m} = \mathbf{F}_{n,m}^\top\mathbf{F}_{n,m}$ and the projection matrix $\mathbf{S}_{n,m} = \mathbf{F}_{n,m}\mathbf{R}_{n,m}^{-1}\mathbf{F}_{n,m}^\top$. We still have that

$$\arg\min_{j=1,\ldots,k}\|\boldsymbol{\phi}_i - \mathbf{c}_{n,m,j}\|^2 = \boldsymbol{\Pi}_m\Phi_n\mathbf{S}_{n,m}\mathbf{e}_i,$$

due to the optimality of $\mathbf{F}_{n,m}$ w.r.t. $\boldsymbol{\Pi}_m\Phi_n$. Substituting in the definition of $W(\cdot,\cdot)$

$$
\begin{aligned}
W(\mathbf{C}_{n,m},\mu_n) &= \frac{1}{n}\sum_{i=1}^{n}\min_{j=1,\ldots,k}\|\boldsymbol{\phi}_i - \mathbf{c}_{n,m,j}\|^2\\
&= \frac{1}{n}\sum_{i=1}^{n}\|\boldsymbol{\phi}_i - \boldsymbol{\Pi}_m\Phi_n\mathbf{S}_{n,m}\mathbf{e}_i\|^2\\
&= \frac{1}{n}\|\Phi_n - \boldsymbol{\Pi}_m\Phi_n\mathbf{S}_{n,m}\|_F^2\\
&= \frac{1}{n}\mathrm{Tr}(\Phi_n^\top\Phi_n - 2\Phi_n^\top\boldsymbol{\Pi}_m\Phi_n\mathbf{S}_{n,m} + \mathbf{S}_{n,m}\Phi_n^\top\boldsymbol{\Pi}_m\boldsymbol{\Pi}_m\Phi_n\mathbf{S}_{n,m}).
\end{aligned}
$$

Since $\mathbf{S}_{n,m}$ is a projection matrix
$$\mathrm{Tr}(\Phi_n^\mathsf{T}\mathbf{\Pi}_m\Phi_n\mathbf{S}_{n,m}) = \mathrm{Tr}(\Phi_n^\mathsf{T}\mathbf{\Pi}_m\Phi_n\mathbf{S}_{n,m}\mathbf{S}_{n,m})$$
$$= \mathrm{Tr}(\mathbf{S}_{n,m}\Phi_n^\mathsf{T}\mathbf{\Pi}_m\Phi_n\mathbf{S}_{n,m}),$$

and using the optimality of $\mathbf{F}_{n,m}$ we have

$$\|\Phi_n - \mathbf{\Pi}_m\Phi_n\mathbf{S}_{n,m}\|_F^2 = \mathrm{Tr}(\Phi_n^\mathsf{T}\Phi_n - \mathbf{S}_{n,m}\Phi_n^\mathsf{T}\mathbf{\Pi}_m\Phi_n\mathbf{S}_{n,m})$$
$$= \mathrm{Tr}(\Phi_n^\mathsf{T}\mathbf{\Pi}_m\Phi_n - \mathbf{S}_{n,m}\Phi_n^\mathsf{T}\mathbf{\Pi}_m\Phi_n\mathbf{S}_{n,m}) + \mathrm{Tr}(\Phi_n^\mathsf{T}\Phi_n - \Phi_n^\mathsf{T}\mathbf{\Pi}_m\Phi_n)$$
$$\leq \mathrm{Tr}(\Phi_n^\mathsf{T}\mathbf{\Pi}_m\Phi_n - \mathbf{S}_n\Phi_n^\mathsf{T}\mathbf{\Pi}_m\Phi_n\mathbf{S}_n) + \mathrm{Tr}(\Phi_n^\mathsf{T}\Phi_n - \Phi_n^\mathsf{T}\mathbf{\Pi}_m\Phi_n)$$
$$= \mathrm{Tr}(\Phi_n^\mathsf{T}\Phi_n - \mathbf{S}_n\Phi_n^\mathsf{T}\mathbf{\Pi}_m\Phi_n\mathbf{S}_n).$$

Using Definition 2 we have $\mathbf{\Pi}_m - \mathbf{\Pi}_n \succeq -\frac{\gamma}{1-\varepsilon}(\Phi_n\Phi_n^\mathsf{T} + \gamma\mathbf{\Pi}_n)^{-1}$ and

$$\mathrm{Tr}(\Phi_n^\mathsf{T}\Phi_n - \mathbf{S}_n\Phi_n^\mathsf{T}\mathbf{\Pi}_m\Phi_n\mathbf{S}_n)$$
$$\leq \mathrm{Tr}\left(\Phi_n^\mathsf{T}\Phi_n - \mathbf{S}_n\Phi_n^\mathsf{T}\Phi_n\mathbf{S}_n + \frac{\gamma}{1-\varepsilon}\mathbf{S}_n\Phi_n^\mathsf{T}(\Phi_n\Phi_n^\mathsf{T} + \gamma\mathbf{\Pi}_n)^{-1}\Phi_n\mathbf{S}_n\right)$$
$$\leq \mathrm{Tr}\left(\Phi_n^\mathsf{T}\Phi_n - \mathbf{S}_n\Phi_n^\mathsf{T}\Phi_n\mathbf{S}_n + \frac{\gamma}{1-\varepsilon}\mathbf{S}_n\Phi_n^\mathsf{T}(\Phi_n\Phi_n^\mathsf{T})^+\Phi_n\mathbf{S}_n\right)$$
$$= W(\mathbf{C}_n, \mu_n) + \frac{\gamma}{1-\varepsilon}\mathrm{Tr}\left(\mathbf{S}_n\Phi_n^\mathsf{T}(\Phi_n\Phi_n^\mathsf{T})^+\Phi_n\mathbf{S}_n\right).$$

Noting now that $\|\Phi_n^\mathsf{T}(\Phi_n\Phi_n^\mathsf{T})^+\Phi_n\| \leq 1$, $\mathbf{S}_n$ is a projection matrix, and $\mathrm{Tr}(\mathbf{S}_n) = k$ we have

$$\frac{\gamma}{1-\varepsilon}\mathrm{Tr}\left(\mathbf{S}_n\Phi_n^\mathsf{T}(\Phi_n\Phi_n^\mathsf{T})^+\Phi_n\mathbf{S}_n\right) \leq \frac{\gamma}{1-\varepsilon}\mathrm{Tr}\left(\mathbf{S}_n\mathbf{S}_n\right) = \frac{\gamma k}{1-\varepsilon}.$$

Conversely, if we focus on the projection matrix $\Phi_n^\mathsf{T}(\Phi_n\Phi_n^\mathsf{T})^+\Phi_n = \Phi_n^\mathsf{T}\Phi_n(\Phi_n\Phi_n^\mathsf{T})^+ = \mathbf{K}_n\mathbf{K}_n^+$, and letting $r = \mathrm{Rank}(\mathbf{K}_n)$ we have

$$\frac{\gamma}{1-\varepsilon}\mathrm{Tr}\left(\mathbf{S}_n\Phi_n^\mathsf{T}(\Phi_n\Phi_n^\mathsf{T})^+\Phi_n\mathbf{S}_n\right) \leq \frac{\gamma}{1-\varepsilon}\mathrm{Tr}\left(\Phi_n^\mathsf{T}(\Phi_n\Phi_n^\mathsf{T})^+\Phi_n\right) \leq \frac{\gamma r}{1-\varepsilon}.$$

Since both bounds hold simultaneously, we can simply take the minimum to conclude our proof. $\square$

*Proof of Theorem 2.* Given our dictionary $\mathcal{I}$, we need to change our decomposition. We denote with $\mathbb{E}_\mathcal{A}[W(\mathbf{C}_{n,m}^{++}, \mu)]$ the expectation over the randomness of the $k$-means++ seeding and Loyd algorithm. Then

$$\mathop{\mathbb{E}}_{\mathcal{D}\sim\mu}\left[\mathop{\mathbb{E}}_\mathcal{A}[W(\mathbf{C}_{n,m}^{++}, \mu)]\right] = \mathop{\mathbb{E}}_{\mathcal{D}\sim\mu}\left[\mathop{\mathbb{E}}_\mathcal{A}[W(\mathbf{C}_{n,m}^{++}, \mu)] - W(\mathbf{C}_{n,m}^{++}, \mu_n)\right] + \mathop{\mathbb{E}}_{\mathcal{D}\sim\mu}\left[\mathop{\mathbb{E}}_\mathcal{A}[W(\mathbf{C}_{n,m}^{++}, \mu_n)]\right].$$

Once again the first term can be bounded as $\mathcal{O}(k/\sqrt{n})$ using the stronger [10, Lemma 4.3], so we turn our attention on the second term. From Proposition 3 we have

$$\mathop{\mathbb{E}}_{\mathcal{D}\sim\mu}\left[\mathop{\mathbb{E}}_\mathcal{A}[W(\mathbf{C}_{n,m}^{++}, \mu_n)]\right] \leq \mathop{\mathbb{E}}_{\mathcal{D}\sim\mu}\left[8(\log(k) + 2)W(\mathbf{C}_{n,m}, \mu_n)\right]$$
$$\leq \mathop{\mathbb{E}}_{\mathcal{D}\sim\mu}\left[8(\log(k) + 2)\left(W(\mathbf{C}_n, \mu_n) + \frac{k}{1-\varepsilon}\frac{\gamma}{n}\right)\right],$$

where we used Lemma 3 in the second inequality. Adding and subtracting $W(\mathbf{C}_m, u)$, using once again Proposition 1, and putting everything together we have

$$\mathop{\mathbb{E}}_{\mathcal{D}\sim\mu}\left[\mathop{\mathbb{E}}_\mathcal{A}[W(\mathbf{C}_{n,m}^{++}, \mu)]\right] \leq \mathcal{O}\left(\frac{k}{\sqrt{n}}\right) + \mathop{\mathbb{E}}_{\mathcal{D}\sim\mu}\left[\mathop{\mathbb{E}}_\mathcal{A}[W(\mathbf{C}_{n,m}^{++}, \mu_n)]\right]$$
$$\leq \mathcal{O}\left(\frac{k}{\sqrt{n}}\right) + 8(\log(k) + 2)\left(\mathop{\mathbb{E}}_{\mathcal{D}\sim\mu}[W(\mathbf{C}_n, \mu_n)] + \frac{k}{1-\varepsilon}\frac{\gamma}{n}\right)$$
$$\leq \mathcal{O}\left(\frac{k}{\sqrt{n}}\right) + 8(\log(k) + 2)\left(\mathcal{O}\left(\frac{k}{\sqrt{n}}\right) + W^*(\mu) + \frac{k}{1-\varepsilon}\frac{\gamma}{n}\right)$$
$$\leq \mathcal{O}\left(\log(k)\left(\frac{k}{\sqrt{n}} + W^*(\mu) + k\frac{\gamma}{n}\right)\right),$$

which concludes our proof. $\square$

*Proof of Lemma 2.* Before starting the proof, we need the following result, which is a trivial extension of [7, Lemma 2] to an RKHS, see also [14].

**Corollary 2** ([7])**.** *Let $\mathcal{I}$ be constructed by uniformly sampling $m \geq 12\kappa^2 n/\gamma \log(n/\delta)/\varepsilon^2$ points from $\Phi_n$ with replacement. Then with probability at least $1 - \delta$ we have*

$$\left\| (\Phi_n \Phi_n^\intercal + \gamma \mathbf{\Pi}_n)^{-1/2} \left( \Phi_n \Phi_n^\intercal - \frac{n}{m} \Phi_m \Phi_m^\intercal \right) (\Phi_n \Phi_n^\intercal + \gamma \mathbf{\Pi}_n)^{-1/2} \right\| \leq \varepsilon,$$

*which implies*

$$(1 - \varepsilon) \Phi_n \Phi_n^\intercal - \varepsilon\gamma \mathbf{\Pi}_n \preceq \frac{n}{m} \Phi_m \Phi_m^\intercal \preceq (1 + \varepsilon) \Phi_n \Phi_n^\intercal + \varepsilon\gamma \mathbf{\Pi}_n$$

Using $\mathbf{\Pi}_n$'s definition

$$\mathbf{\Pi}_n = \left( \frac{n}{m} \Phi_m \Phi_m^\intercal + \varepsilon\gamma \mathbf{\Pi}_n \right)^{-1/2} \left( \frac{n}{m} \Phi_m \Phi_m^\intercal + \varepsilon\gamma \mathbf{\Pi}_n \right) \left( \frac{n}{m} \Phi_m \Phi_m^\intercal + \varepsilon\gamma \mathbf{\Pi}_n \right)^{-1/2}$$

$$= \left( \frac{n}{m} \Phi_m \Phi_m^\intercal + \varepsilon\gamma \mathbf{\Pi}_n \right)^{-1/2} \frac{n}{m} \Phi_m \Phi_m^\intercal \left( \frac{n}{m} \Phi_m \Phi_m^\intercal + \varepsilon\gamma \mathbf{\Pi}_n \right)^{-1/2}$$

$$+ \varepsilon\gamma \left( \frac{n}{m} \Phi_m \Phi_m^\intercal + \varepsilon\gamma \mathbf{\Pi}_n \right)^{-1/2} \mathbf{\Pi} \left( \frac{n}{m} \Phi_m \Phi_m^\intercal + \varepsilon\gamma \mathbf{\Pi}_n \right)^{-1/2}$$

$$= \left( \frac{n}{m} \Phi_m \Phi_m^\intercal + \varepsilon\gamma \mathbf{\Pi}_n \right)^{-1/2} \frac{n}{m} \Phi_m \Phi_m^\intercal \left( \frac{n}{m} \Phi_m \Phi_m^\intercal + \varepsilon\gamma \mathbf{\Pi}_n \right)^{-1/2} + \varepsilon\gamma \left( \frac{n}{m} \Phi_m \Phi_m^\intercal + \varepsilon\gamma \mathbf{\Pi}_n \right)^{-1}$$

$$= \left( \frac{n}{m} \Phi_m \Phi_m^\intercal + \varepsilon\gamma \mathbf{\Pi}_n \right)^{-1/2} \frac{n}{m} \Phi_m \Phi_m^\intercal \left( \frac{n}{m} \Phi_m \Phi_m^\intercal + \varepsilon\gamma \mathbf{\Pi}_n \right)^{-1/2} + \varepsilon\gamma \left( \frac{n}{m} \Phi_m \Phi_m^\intercal + \varepsilon\gamma \mathbf{\Pi}_n \right)^{-1} .$$

Note now that

$$\left( \frac{n}{m} \Phi_m \Phi_m^\intercal + \varepsilon\gamma \mathbf{\Pi}_n \right)^{-1/2} \frac{n}{m} \Phi_m \Phi_m^\intercal \left( \frac{n}{m} \Phi_m \Phi_m^\intercal + \varepsilon\gamma \mathbf{\Pi}_n \right)^{-1/2}$$

$$= \frac{n}{m} \Phi_m \left( \frac{n}{m} \Phi_m \Phi_m^\intercal + \varepsilon\gamma \mathbf{\Pi}_n \right)^{-1} \Phi_m^\intercal = \Phi_m \left( \Phi_m \Phi_m^\intercal + \frac{m}{n} \varepsilon\gamma \mathbf{\Pi}_n \right)^{-1} \Phi_m^\intercal$$

$$\preceq \Phi_m \left( \Phi_m \Phi_m^\intercal \right)^+ \Phi_m^\intercal = \mathbf{\Pi}_m$$

And that using Corollary 2 we have

$$\left( \frac{n}{m} \Phi_m \Phi_m^\intercal + \gamma \mathbf{\Pi}_n \right)^{-1} \preceq \left( (1 - \varepsilon) \Phi_n \Phi_n^\intercal + (\varepsilon\gamma - \varepsilon\gamma) \mathbf{\Pi}_n \right)^{-1}$$

$$= \frac{1}{1 - \varepsilon} \left( \Phi_n \Phi_n^\intercal + \frac{\varepsilon\gamma - \varepsilon\gamma}{1 - \varepsilon} \mathbf{\Pi}_n \right)^{-1} = \frac{\gamma}{1 - \varepsilon} \left( \Phi_n \Phi_n^\intercal \right)^+ .$$

Combining these two results we obtain the proof. $\qquad\square$