[Reviews · NeurIPS 2018]

Reviewer 1



The paper addresses the question of finding the number of landmark points for Nystrom approximation when used with Kernel K-means. The main contribution of this paper is to show that \sqrt{n} number of landmarks suffice to get a low regret of O(k/\sqrt{n}). While there has been long line of works in understanding Nystorm methods, the result obtained in this paper is new. It is an interesting development and should be of interest to community involved in Clustering with Kernels. In summary this paper is well written with original results which will be of interest to experts interested in Kernel methods.

Reviewer 2



Summary The paper investigates the kernel k-means problem, proposing a new approximation of the method based on Nystrom embeddings. In particular, both the statistical accuracy and computational efficiency of the proposed approximation is studied. The main contribution is the derivation of a theoretical bound limiting the error cost; the proposed method can achieve such theoretical guarantee with a computational cost of (O(\sqrt{n})). In practice, this implies that using \sqrt{n} points (with n being the original sample size) is enough to obtain a good enough approximation. The authors also propose an approach to choose the points while ensuring a balance trade off between preserving a good approximation and a small complexity (dictionary size). Novelty and Clarity The theoretical results are interesting and provide new insights into the topic. The writing style is easy to follow and the mathematical notation is also clear. Overall opinion The paper is technically correct and provides a new contribution to the field. The experimental section could be improved and some points should be clarified. The submission would be at a good level if the points below are clarified. Major and minor comments are provided below Major comments - Proof of Lemma 1 and Lemma 2 could be provided in the supplementary (or at least a sketch) - In the experiments it is not clear how the parameters, including the number of clusters, are selected. - It would be beneficial to provide more detailed information on the runtime to further support the benefit of performing the approximation - It would be beneficial to provide a more exhaustive comparison (and corresponding detailed information) with the competitors. This seems to be only slightly mentioned for the MNIST8M experiment Minor comments - The acronyms JL and RLS seem to be used without being introduced - Section 4: The Lloyd’s algorithm reference is missing - Algorithm 1 can be reformulated to be presented in a more structured form - Figure 2: Legends and titles as well as a short caption could be added

Reviewer 3



This paper considers the computational and statistical trade off for the kernel K-mean via the Nyström approach, which aims at reducing the computational cost and storage space. This paper proves some surprising results, maintaining optimal rates while the computational cost can be reduced greatly under some basic assumptions. The derived results under the framework of unsupervised learning is very meaning, while some similar results concerning the balance of statistical error and computational accuracy have been established in the supervised setting.